# Clinical Characteristics of Lower-Limb Ischemia in Japanese Patients with Type 2 Diabetes and Usefulness of the Great Toe Blood Flow as a Predictive Indicator of Leg Arterial Obstruction

**DOI:** 10.3390/healthcare10091753

**Published:** 2022-09-13

**Authors:** Aya Sakamoto, Mitsunori Ikeda

**Affiliations:** 1Faculty of Nursing, University of Kochi, Kochi 781-8515, Japan; 2Graduate School of Nursing, University of Kochi, Kochi 781-8515, Japan; 3Wellness and Longevity Center, University of Kochi, Kochi 781-8515, Japan

**Keywords:** blood flow disorder, diabetic foot care, physiological assessment, toe blood flow, type 2 diabetic patients

## Abstract

Nurses are required to make quantitative, evidence-based observations when implementing diabetic foot care. This study aimed to clarify the characteristics of lower-limb ischemia in patients with type 2 diabetes using subjective and objective symptoms and physiological indicators and whether the physiological characteristics are established as predictive indicators of arterial obstruction. Fifty Japanese patients with type 2 diabetes (100 limbs) were classified into three groups using the ankle–brachial index (ABI). Patients with an ABI of ≤0.69 had subjective and objective symptoms of blood flow disturbance, such as pain at rest, cold sensation, pale skin, and imperceptibility to the dorsalis pedis artery. Blood flow in the first toe was the lowest. Binary logistic regression analysis established hallux perfusion as a predictive model for lower-limb arterial occlusion (odds ratio = 0.979, 95% confidence interval 0.900–0.999). Thus, when nurses perform diabetic foot care, it is necessary to evaluate not only subjective and objective symptoms, but also blood flow at the microcirculatory level of the great toe.

## 1. Introduction

The number of patients with diabetes in Japan is increasing rapidly due to the changes in lifestyle and social environment. It has reached a record high of 3,289,000 according to a 2017 survey [1]. Diabetes mellitus is not curable but causes complications such as neuropathy, retinopathy, and nephropathy. Diabetic foot, which is caused by neuropathy, progression of microangiopathy, complications of peripheral artery disease (PAD), and leukocyte dysfunction, has become a crucial problem for patients with diabetes. The global prevalence of diabetic foot ulcer was 6.3%, which was higher in men (4.5%) than in women (3.5%) and higher in patients with type 2 diabetes (T2D) (6.4%) than in those with type 1 diabetes (5.5%) [2]. Once a foot lesion develops, the wound barely heals and progresses to a foot ulcer/gangrene, which becomes more severe when complicated with bacterial infection. Worsening diabetic foot is often amputated [3], resulting in the significant reduction of activities of daily living, quality of life, and poor life prognosis. Patients with diabetes are 7–30 times more susceptible to leg amputations than the general population, accounting for more than half of all amputations [4]. Approximately 50% of people with diabetic ulcers have PAD [5]. The risks for leg amputation are the combination of peripheral neuropathy, diabetes-induced infections, and PAD [6]. Diabetic ulcers are associated with motor, sensory, and autonomic neuropathy, consequent foot deformities, and wound-healing disorders, and these complications are also potential risks for amputation in diabetic foot [6].

To prevent diabetic foot lesions, nurses provide “diabetic foot care”, including foot observation, assessment, nail and toe treatment, lifestyle counseling, and lifestyle guidance to patients with diabetes and their families. The diabetes complication management fee was introduced in the National Health Insurance System in Japan in 2008, and “diabetic foot care” [7] came down to nurses. Specifically, to prevent aggravation, it is necessary to select high-risk patients with diabetic foot lesions and introduce them to early interventions. Thus, developing evidence-based systematic observations with quantitative methods is important. However, assessment methods commonly used in nursing are non-quantifiable ones, including visual and manual sensations and patient surface symptoms, such as skin temperature, skin color tone, arterial palpation, and sensory tests using simple instruments [8]. These observations largely rely on the evaluators’ experience. The usefulness of transcutaneous measurement of oxygen tension (TcPO_2_) [9] and the amount of melanin pigment [10] have been reported as physiological indicators that can be used by nurses for assessing lower-limb arterial obstruction. However, a few papers have argued the relationship of these physiological indicators with subjective and objective symptoms. The ankle–brachial index (ABI) [9] is also used in nursing practice to evaluate lower-limb blood flow disorders. In the present study, we performed ABI measurement to estimate the degree of lower-limb obstruction in patients with T2D by the Doppler method. Then, we assessed the symptoms of arterial obstruction in the diabetic foot by means of subjective symptoms, objective symptoms, and physiological indicators such as toe blood flow using the laser blood flow meter, acceleration pulse wave aging index, and TcPO_2_ of the dorsal foot. Furthermore, we examined whether these physiological indicators were useful in the prediction of arterial obstruction.

## 2. Materials and Methods

### 2.1. Subjects

The participants were patients with T2D aged ≥ 65 years who were attending or were hospitalized for diabetes treatment at two facilities located in our district. Fifty participants (30 men and 20 women) who consented to participate in the study during the data collection period were included in the study. Those with a history of toe or lower-limb amputation and those who were regularly treated with hemodialysis were excluded. Lower limbs (100 limbs) were examined in the study.

We confirmed the participants’ willingness to cooperate in the study after thoroughly explaining the details of the study, including its significance, ethical considerations for participation, assurance of their free will to participate in the study, assurance of their freedom to withdraw their participation, benefits and disadvantages of participating in the study, risks to the individual arising from the research activities, publication of the results in a reputable journal, and expected contribution to the research field. We asked permission to access laboratory data from participants’ medical records. Signed informed consent was obtained from the participants. In addition, the study was reviewed and approved by the Research Ethics Review Committee of University of Kochi (Approval No. 18–48) before the start of the study, and it adhered to the Declaration of Helsinki.

### 2.2. Data Collection

We collected data on foot ischemia based on participants’ subjective symptoms, researcher’s objective symptoms, and physiological indices. Subjective symptoms included cold feeling, numbness, and pain at rest in each lower limb. Objective symptoms obtained were sensory disturbance, presence of coldness, skin pallor, skin dryness, and palpation of the dorsalis pedis and posterior tibial arteries of each foot. Three physiological indices were employed: blood flow using the laser blood flowmeter, the accelerated pulse wave aging index, and TcPO_2_. An aneroid sphygmomanometer and ultrasound blood flow meter were used to measure ABI for assessing blood flow disturbances in the lower limbs. Laboratory data were collected from medical records, and information on pre-existing diseases and treatment was collected by history taking. The researcher conducted several pretests to familiarize oneself with the operation of the devices before implementation. Data collection was conducted by a single researcher, and measurements took approximately 60 min to complete. The data collection period was from March to October 2019.

### 2.3. Physiological Measurement

#### 2.3.1. ABI Value

ABI values were measured using the ultrasonic Doppler method with a small aneroid sphygmomanometer and an ultrasonic blood flow meter. The ABI for each leg was calculated by the ratio of the higher ankle pressure (dorsal pedis artery or posterior tibial artery) over the higher of the two arm pressures. For ankle systolic pressure, the higher pressures of the dorsal pedis artery and posterior tibial artery were used. The Doppler method measuring ABI has gained reliability in previous studies [11,12].

#### 2.3.2. Toe Blood Flow

Toe blood flow was measured using a laser blood flowmeter pocket LDF (JMS Corporation, Hiroshima, Japan) according to the instructions of the manufacturer. The measurements sites were the pads of the great to the fifth toes. The probes were taped to the right and left measurement sites. While the measurement standby screen was displayed, a snapshot function was used to conduct three measurements simultaneously on the right and left sides, and the average value was used as the blood flow of each toe.

#### 2.3.3. Acceleration Pulse Wave Aging Index

The acceleration pulse wave was measured using the acceleration pulse wave measurement system ARTET CDN (U-Medica Corporation, Osaka, Japan), and the acceleration pulse wave aging index was calculated from the measured values. The sensor was placed on the great toe and secured with tape, and the acceleration pulse wave measurement was performed for 150 pulsebeats. The acceleration pulse wave aging index was calculated as follows: b–c–d–e/a, where a, b, c, d, and e are five vertices of the acceleration pulse wave at points a, b, c, d, and e, respectively [13].

#### 2.3.4. TcPO_2_ of the Dorsal Foot

TcPO_2_ was measured using the PO-850A transcutaneous O_2_/CO_2_ gas monitor (Shinsei Electronics Co., Ltd., Amagasaki, Japan). The measurement probe was attached via a special apparatus on the skin of the dorsal side of the metatarsal bone (dorsal side of the foot 2.5 cm above the base of the middle of the great and second toes), immediately avoiding the site above the bone and the route of the superficial vein. After 15 min of supine rest, the right and left TcPO_2_ were measured in the same position.

### 2.4. Examination Procedure

The temperature in the room where measurements were obtained was set at 25–28 °C, and locations near windows, in direct sunlight, or directly exposed to the wind of an air conditioner were avoided. The room was also kept quiet and free from loud noises.

Participants were asked whether they had smoked within 1 h of the measurement or had consumed caffeine and were asked about the partial tightness of their clothing. After hearing participants’ subjective foot symptoms, socks, stockings, etc., were removed, and the participant was placed supine on the bed. After 15 min of rest, vital signs and ankle blood pressure were measured. The participants were then examined for the appearance of the right and left feet, and the dorsalis pedis and posterior tibial arteries were palpated. Toe blood flow, TcPO_2_, and acceleration pulse wave aging index were then measured.

### 2.5. Data Analysis

Patients with diabetes were classified according to the severity of arterial obstruction of lower limbs determined by ABI originally used for PAD [14]. Patients with ABI of ≥0.91 was defined as the “normal group”, 0.70–0.90 as the “mild obstruction group”, and ≤0.69 as the “moderate-to-severe obstruction group”. The one-way analysis of variance and the Kruskal–Wallis test were used to compare basic data with physiological indicators. Binomial logistic regression analysis with the forced-entry method was used to predict the development of arterial obstruction. IBM SPSS Statistics, version 25 (IBM Corp., Armonk, NY, USA), was used for statistical analysis, with a significance level of 5%.

## 3. Results

### 3.1. Overview of the Participants

The average age of the 50 participants (20 women and 30 men) was 72.7 (*SD* 4.8) years, with an average diabetes duration of 16.3 (*SD* 12.2) years, and the average glycosylated hemoglobin (HbA1c) and hemoglobin (Hb) levels were 7.3% (*SD* 1.0%) and 13.6 (*SD* 1.5) g/dL, respectively. Of the 100 lower limbs examined, two limbs with ABI values of 1.31 and 1.33 were excluded because of the possibility of vascular calcification; therefore, 98 limbs were included in the analysis.

### 3.2. Comparison of Demographic Data, Diabetic Complications, and Pre-Existing Diseases by the Degree of Lower-Limb Obstruction

The demographic data of the three groups classified by the ABI value are shown in Table 1. No significant differences were found in age (*p* = 0.973), duration of diabetes (*p* = 0.437), HbA1c level (*p* = 0.947), and Hb level (*p* = 0.300) among the three groups.

We compared the presence or absence of diabetic complications and pre-existing diseases in each of the three groups classified by ABI (Table 2). In the moderate-to-severe group, 66.7% of the cases were accompanied with diabetic neuropathy (*p* = 0.001), diabetic retinopathy (*p* < 0.001), and diabetic nephropathy (*p* < 0.001), which are complications of microangiopathy, and arteriosclerosis obliterans (*p* < 0.001), with significant differences. Hyperlipidemia was a less common complication in all groups. Ischemic heart diseases and cerebrovascular diseases were observed in 33.3% of the patients in the moderate-to-severe group, whereas they were observed less in the normal (19.8%) and mild obstruction groups (14.3%).

### 3.3. Comparison of the Presence or Absence of Subjective and Objective Symptoms of Ischemia by the Degree of Lower-Limb Obstruction

Table 3 shows the subjective and objective symptoms of ischemia depending on the degree of obstruction of the lower limbs. Among the subjective symptoms, resting pain was as high as 66.7% in the moderate-to-severe group but not in the normal and mild obstruction groups (*p* < 0.001). Surprisingly, the moderate-to-severe obstruction group did not complain about cold sensation, but the normal (27.2%) and mild obstruction groups did (7.1%).

As for objective symptoms, the proportion of cold sensation and pallor of the foot skin and the unpalpable dorsalis pedis artery accounted for 66.7% in the moderate-to-severe group. Moreover, the posterior tibial artery could not be palpated in all limbs of the moderate-to-severe group. The two limbs with resting pain showed low ABI values of 0.39 and 0.28, accompanied with a feeling of objective cold sensation and pallor of the skin without the palpation of dorsal pedis and posterior tibial arteries (data not shown in Table 3). Statistical differences in cold sensation (*p* = 0.034), pallor of the skin (*p* = 0.025), and unpalpable dorsalis pedis artery (*p* < 0.001) and posterior tibial artery (*p* = 0.002) were found among the three groups.

### 3.4. Comparison of Physiological Indicators by the Degree of Lower-Limb Obstruction

Toe blood flow, acceleration pulse wave aging index, and TcPO_2_ of the participants’ limbs were measured (Table 4). A significant difference was found in the toe blood flow of the great (*p* = 0.034) and second (*p* = 0.048) toes among the three groups. The great toe showed increased blood flow in the mild obstruction group (58.5 mL/min) and decreased blood flow in the moderate-to-severe obstruction group (14.8 mL/min) compared with the normal group (33.4 mL/min). The second toe showed increased blood flow in both the mild (41.7 mL/min) and moderate-to-severe obstruction groups (39.5 mL/min) compared with the normal group (26.4 mL/min). A significant difference was found in the blood flow of the third to the fifth toes among the three groups.

The acceleration pulse wave aging index was not calculated in two limbs of the moderate-to-severe obstruction group and one limb of the mild obstruction group because the acceleration pulse wave could not be detected. Only one value was listed as the datum of the acceleration pulse wave aging index of the moderate-to-severe obstruction group. Therefore, statistical analysis among the three groups could not be performed. However, the absolute value of the acceleration pulse wave aging index of the mild obstruction group (−0.18) was lower than that of the normal group (−0.25), indicating a more aged type of blood vessels of the mild obstruction group.

TcPO_2_ of all groups was approximately 50 mmHg. Statistical analysis showed no significant difference in the TcPO_2_ level (*p* = 0.450), indicating almost the same levels of O_2_ supply in the toes of all groups.

### 3.5. Prediction of Lower-Limb Arterial Obstruction by the Physiological Index

Since the great toe blood flow was significantly different in the comparison of the three groups (Figure 1, Figure 2 and Figure 3), the great toe blood flow might indicate ABI severity. To clarify the physiological indices that can predict the presence of arterial obstruction, the presence or absence of arterial obstruction was used as a dependent variable, and the blood flow in the great toe as an independent variable (Table 4). A binomial logistic regression analysis was performed by the forced-injection method (odds ratio 0.979, 95% confidence interval 0.900–0.999).

The result of model χ^2^ was *p* = 0.035, and a model of significant variables was established (Table 4). The overall prediction accuracy was 82.7%. As a result, the physiological index that could predict the presence or absence of arterial obstruction in patients with T2D was the great toe blood flow.

## 4. Discussion

In our investigation, although old age, long-term diabetic history, and increment of HbA1c were detected, the degree of lower-limb obstruction was classified into the normal group, mild obstruction group, and moderate-to-severe obstruction group. Such demographic data did not show significant differences among the three groups. On the contrary, in the moderate-to-severe obstruction group, the ratios of diabetic complications, such as neuropathy, retinopathy, and nephropathy, were high. The ratios of arteriosclerosis obliterans, a macroangiopathy, and ischemic heart disease/cerebrovascular disease were also high. Thus, lesions of both small and large blood vessels progressed systemically when the degree of lower-limb obstruction was high.

Impaired cold sensation and worsening of resting pain were subjective symptoms in the moderate-to-severe obstruction group. This group barely complained of numbness. Contrary to subjective symptoms, objective symptoms include cold sensation by palpation and skin paleness by inspection. The cold pain threshold of patients with diabetes increased compared with healthy individuals [15]. Therefore, patients with T2D may be subjectively insensitive to cold sensation via an increased threshold. Tingling pain and numbness (reduced ability to feel pain) are paradoxical sensorial impairments, but both belong to the distal symmetric peripheral neuropathy. They were reported as the most common complaints [16]. Our observations suggest that mild nerve damage increases numbness, but severe damage increases resting pain. Skin pallor is caused by reduced blood flow and reduced oxygen content. Leg elevation sometimes worsens skin pallor. An increase in skin pallor, in accordance with an increased objective cold sensation, indicates less supply of arterial blood, bringing about both a high O_2_ concentration and core body temperature.

The palpation of the dorsal pedis artery has been generally used as a simple method for assessing the condition of peripheral circulation. Several participants in the normal group had a non-palpating dorsal pedis artery. The dorsal pedis artery was non-palpable in 24 (2.2%) of 1094 feet of healthy individuals examined by Doppler examination [17], which is due to an anatomical variant of anterior tibial artery bifurcation at the ankle level [18], resulting in the absence of the dorsalis pedis artery in the dorsal foot. We have to consider this variant in palpating the dorsal pedis artery.

In this study, the blood flow in the great toe was decreased in the moderate-to-severe obstruction group. Since the great toe plays an important role in bearing the full body weight in the standing position, a loss of the great toe disturbs patients’ daily living activity. Great toe amputation contributes to the development of deformities of the second and third toes and lesser metatarsophalangeal joints and new ulcer formation in patients with diabetes [19]. The blood flow in the second toe was revealed to be significantly different among the groups classified by ABI severity. However, disturbed blood flow did not denote the same tendency of the degree of vascular obstruction: increased blood flow in the mild obstruction group but decreased again in the moderate-to-severe obstruction group. Even though a significant difference was found the pathophysiological significance remains unclear. Although a low blood flow in the great toe was found in the moderate-to-severe obstruction group, the blood flow measurement by the laser blood flow meter varied by toes, suggesting the necessity of the blood flow measurement of each toe in the foot care for T2D.

The acceleration pulse wave aging index could not be measured in two of three toes of the moderate-to-severe obstruction group. Despite the lower absolute value of the aging index in the mild obstruction group than that of the normal group, this physiological examination should be applied not to the foot with severely disturbed blood flow but to the foot with largely sufficient blood flow.

According to the Wound, Ischemia, and Foot Infection classification, non-ischemic foot is defined as a TcPO_2_ value > 60 mmHg and mild ischemic foot as a TcPO_2_ value of 40–59 mmHg [20]. All our study groups had 50.0–52.5 mmHg of TcPO_2_, indicating mild ischemia. In addition, the difference in TcPO_2_ was not detected among the three groups. Fejfarová et al. [21] performed a similar study using a transcutaneous O_2_/CO_2_ gas monitor against 107 diabetic patients with mild-to-moderate PAD and reported a mild ischemic foot with a TcPO_2_ value of 41.0 mmHg. They also employed TcPO_2_ measurement after the modified Ratschow stimulation test to elucidate an impairment of lower-leg blood flow, including latent PAD and restenosis/obliteration. The Ratschow test included elevating the lower limbs (30 cm above the bed), followed by rhythmic maximal plantarflexion and extension of the talocrural joints for 2 min to decrease O_2_ supply and increase O_2_ consumption of the limbs. The percentage of TcPO_2_ reduction after the modified Ratschow stimulation test was 30.7%, and the duration of TcPO_2_ recovery was 4415 s [21]. The sensitivity of resting TcPO_2_ was very low for PAD diagnosis; however, when the TcPO_2_ measurement was augmented by the modified Ratschow test, the sensitivity and specificity of all TcPO_2_ stimulation parameters increased significantly by up to 25% and 11%, respectively [21]. We verified the significance of TcPO_2_ after the modified Ratschow stimulation, however the Ratschow stimulation is difficult to use in nursing science and daily clinical practice because the test carries a high risk to aggravate the leg ischemia.

In the measurement of the toe blood flow using a laser blood flowmeter pocket LDF, the blood flow in the great toe deteriorated in accordance with the severity of lower-limb arterial obstruction or ABI. Therefore, we verified whether the blood flow of the great toe is a predictor of lower-limb arterial obstruction and found that the blood flow of the great toe predicts the presence or absence of lower-limb arterial obstruction. The reason why the blood flow of the great toe alone is a predictive indicator of the lower-limb arterial obstruction appears to be anatomical arrangement of human leg arteries. The main blood flow to the great toe comes from the dorsal pedis artery, a direct branch of the anterior tibial artery, in the dorsal foot; furthermore, it comes from the medial plantar artery, a branch of the posterior tibial artery, in the sole. It is plausible to think that the blood flow of the great toe reflects the flow of the main anterior/posterior tibial arteries located directory upstream.

Patients with diabetes should aim to avoid the amputation of not only lower legs but also foot toes. The measurement of blood flow in every toe of patients with T2D using a laser blood flowmeter can be easily employed in the foot care practice. The advantages of toe blood flow measurement are as follows: acquisition of the fluid dynamic information of the digital arteries flowing into each toe, acquisition of the blood flow of the great toe as an aggravation-related indicator, and acquisition of the blood flow of the great toe as a predictor of ABI. Ubbink et al. [22] also stated that microcirculatory parameters such as toe blood flow are a useful addition to standard microcirculatory techniques of assessing the severity of lower-limb ischemia, particularly in patients in whom microcirculatory parameters are unattainable. Even if the ABI, which indicates blood flow disturbance in large blood vessels, is normal, foot ulcers will inevitably occur if there is blood flow disturbance in the small blood vessels in the toes, which are more peripheral to the foot. Thus, rather than evaluating blood flow disturbance in the lower limbs only by ABI, which indicates blood flow disturbance in the great vessels, the blood flow in the toes, which is more peripheral, is quantitatively evaluated along with subjective and objective symptoms. Evaluation is important and grasping the blood flow in the great toe is useful for predicting blood flow disorders in patients with T2D.

## 5. Limitations of This Study

The results derived from this study were obtained from a cross-sectional analysis. They were not validated longitudinally. Moreover, there could be a bias in diabetes control among the participants as it cannot be said that past medication history did not influence the study results. In addition, this study could not compare the results with the degree of vascular stenosis or occlusion as determined by angiography, CT, or other imaging studies.

## 6. Conclusions

In this study, we used subjective and other symptoms, as well as physiological indices, to characterize blood flow disturbances in the lower limbs of patients with T2D and to investigate the possibility that these physiological features can be established as predictive indices of blood flow disturbances. The following conclusions were reached:Subjective symptoms of impaired blood flow and other symptoms such as pain at rest, cold sensation, skin pallor, and unpalpable dorsal foot artery were required when the ABI was ≤0.69. Toe blood flow was the lowest in the great toe, but the accelerated pulse wave aging index suggested progressive arteriosclerosis, and TcpO_2_ showed no characteristic values.Binomial logistic regression analysis was performed on the relationship between the presence of impaired blood flow in the lower limbs and physiological indices in patients with T2D, and the great toe blood flow by pocket LDF was established as predictive of impaired blood flow in the lower limbs.

## Figures and Tables

**Figure 1 healthcare-10-01753-f001:**
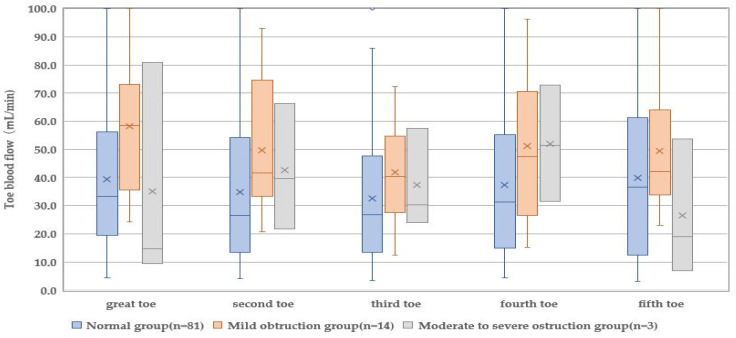
Comparison of toe blood flow according to the degree of obstruction in the lower extremities.

**Figure 2 healthcare-10-01753-f002:**
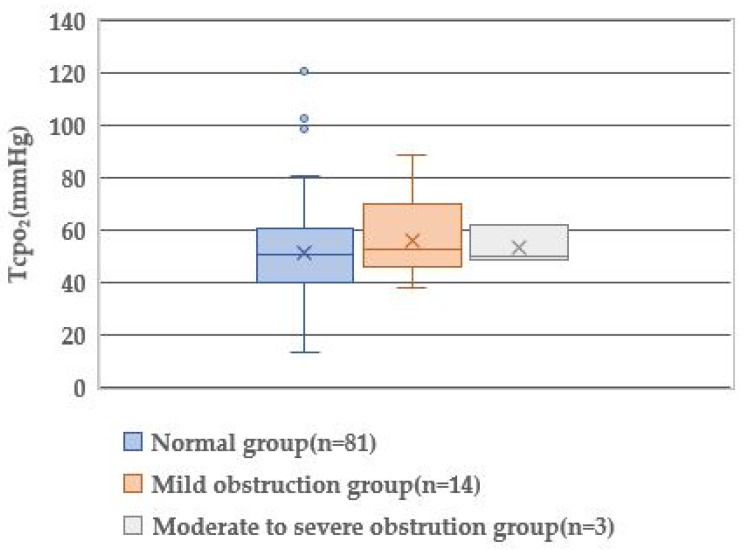
Comparison of TcpO_2_ by degree of lower extremity obstruction.

**Figure 3 healthcare-10-01753-f003:**
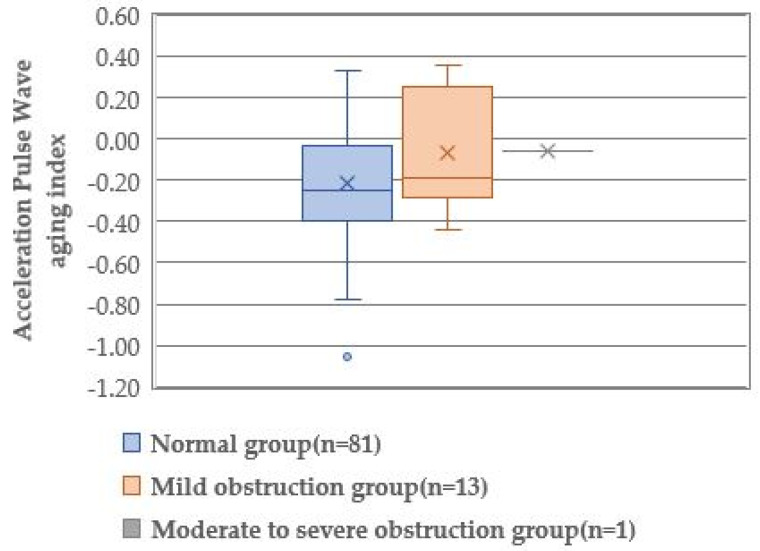
Comparison of acceleration pulse wave aging index by degree of obstruction of lower extremities.

**Table 1 healthcare-10-01753-t001:** Comparison of demographic data by the degree of lower-limb obstruction (*n*
*=* 98).

Variable	Normal Group (*n =* 81)	Mild Obstruction Group (*n =* 14)	Moderate-to-Severe Obstruction Group (*n =* 3)	*p ^a^*
Age (years)	73.0 (65.0–83.0)	73.0 (68.0–76.0)	73.0 (68.0–75.0)	0.973
Diabetes history (years)	13 (1.0–49.0)	20.0 (13.0–37.0)	20.0 (12.3–13.0)	0.437
HbA1c (%)	7.1 (5.3–9.8)	7.1 (6.7–7.4)	7.1 (6.7–7.4)	0.949
Hb (g/dL)	13.6 (11.3–17.8)	12.3 (11.3–14.2)	12.3 (11.3–14.2)	0.300

Note. Data are presented as median (interquartile range). HbA1c, glycosylated hemoglobin; Hb, hemoglobin. *^a^* Kruskal–Wallis test.

**Table 2 healthcare-10-01753-t002:** Diabetic complications and pre-existing diseases by the degree of lower-limb obstruction (*n* = 98).

Variable	Normal Group (*n =* 81)		Mild Obstruction Group (*n =* 14)		Moderate-to-Severe Obstruction Group (*n =* 3)		*p ^a^*
	Absent (%)	Present (%)	Absent (%)	Present (%)	Absent (%)	Present (%)	
Diabetes neuropathy	74 (91.4)	7 (8.6)	14 (100.0)	0 (0.0)	1 (33.3)	2 (66.7)	0.001
Diabetic retinopathy	79 (97.5)	2 (2.5)	11 (78.6)	3 (21.4)	1 (33.3)	2 (66.7)	<0.001
Diabetic nephropathy	81 (100.0)	0 (0.0)	14 (100.0)	0 (0.0)	1 (33.3)	2 (66.7)	<0.001
Hypertension	48 (60.8)	31 (39.2)	4 (28.6)	10 (71.4)	2 (66.7)	1 (33.3)	0.077
Hyperlipidemia	71 (89.9)	8 (10.1)	13 (92.9)	1 (7.1)	3 (100.0)	0 (0.0)	0.806
Arterioscleosis obliterans	81(100.0)	0 (0.0)	14 (100.0)	0 (0.0)	1 (33.3)	2 (66.7)	<0.001
Ischemic heart disease/cerebrovascular disease	65 (80.2)	16 (19.8)	12 (85.7)	2 (14.3)	2(66.7)	1 (33.3)	0.074

Note. *^a^* Centralized distribution analysis.

**Table 3 healthcare-10-01753-t003:** Comparison of the presence or absence of subjective and objective symptoms of ischemia depending on the degree of lower-limb obstruction (*n* = 98).

Variable	Normal Group (*n =* 81)		Mild Obstruction Group (*n =* 14)		Moderate-to-Severe Obstruction Group (*n =* 3)		*p* ^a^
	Absent (%)	Present (%)	Absent (%)	Present (%)	Absent (%)	Present (%)	
Subjective symptom							
Cold sensation	59 (72.8)	22 (27.2)	13 (92.9)	1 (7.1)	3 (100.0)	0 (0.0)	0.168
Numbness	63 (77.8)	18 (22.2)	11 (78.6)	3 (21.4)	3 (100.0)	0 (0.0)	0.008
Resting pain	81 (100.0)	0 (0.0)	14 (100.0)	0 (0.0)	1 (33.3)	2 (66.7)	<0.001
Objective symptom							
Cold sensation	49 (60.5)	32 (39.5)	13 (92.9)	1 (7.1)	1 (33.3)	2 (66.7)	0.034
Pallor	70 (86.4)	11 (13.6)	13 (92.9)	1 (7.1)	1 (33.3)	2 (66.7)	0.025
Dry skin	45 (55.6)	36 (44.4)	5 (35.7)	9 (64.3)	3 (100.0)	0 (0.0)	0.106
Palpation of the dorsalis pedis artery	4 (4.9)	77 (95.1)	2 (14.3)	12 (85.7)	2 (66.7)	1 (33.3)	<0.001
Palpation of the posterior tibial artery	14 (17.3)	67 (82.7)	3 (23.1) ^b^	10 (76.9)	3 (100.0)	0 (0.0)	0.002

Note. *^a^* One-way analysis of variance (ANOVA), ^b^
*n* = 13.

**Table 4 healthcare-10-01753-t004:** Prediction of lower-limb arterial obstruction by the physiological index (*n* = 98).

Variable	Partial Regression Variable	Standard Error	Wald	*p*	Odds Ratio	95% Confidence Interval for Odds Ratio	
						Lower Limit	Upper Limit
Blood flow in the the great toe Variable	−0.021 2.579	0.010 0.586	4.425 19.383	0.035 < 0.001	0.979 13.181	0.900	0.999

Note. A binary logistic regression analysis with the forced entry. Model *χ*^2^
*p* = 0.035, Cox–Snell *R*^2^ = 0.044, Nagelkerke *R*^2^ = 0.074, discriminatory predictive value 82.7%.

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
