# Peer review of "Clinical Characteristics of Lower-Limb Ischemia in Japanese Patients with Type 2 Diabetes and Usefulness of the Great Toe Blood Flow as a Predictive Indicator of Leg Arterial Obstruction"

_healthcare, 2022, doi:10.3390/healthcare10091753_

Round 1

Reviewer 1 Report

Which criteria was taken in selection of sample size

Statistical analysis needs to be elaborate further for the betterment of manuscript.

Conclusion is not proper

Author Response

Comment1 Answer : We have added to the summary what we have drawn from our conclusions about the need for nurses to assess diabetic foot care.

Comment2 Answer : We have checked and corrected the text.

Comment3 Answer : We wrote the "conclusion" in the text.

Reviewer 2 Report

First of all, The authors grouped DMF patients and summarized the comparison of measurements using each subject and object symptoms and physiological indicators. You did a great job organizing the vast amount of data.

The overall design of the article is well thought out.

1) line 75- What is the reason for selecting the subject as 65 years of age or older?

2) On line 163, do the two limbs mean one patient or each patient? 98 limbs would be correct if there is one person, but if there are two people, I think both limbs should be excluded. So the total would have to be 96 limbs. If one person has an abnormality in one place, I think that the other should also be excluded from the study.

3) In table4, the authors' results are well-organized. However, I would like to make a graph to help readers understand.

Author Response

Comment1 Answer : Because the variation in the age of the subjects would increase the number of influencing factors, we chose subjects over 65 years old, which is considered elderly in Japan.

Comment2 Answer : If one leg is expected to be abnormal, it is expected that the other will also become abnormal due to systemic arteriosclerosis, etc. However, since changes in the lower limbs (limbs) are not expected to occur in both legs at the same time, only the abnormal leg was excluded in this study.

Comment3 Answer : I bluffed the results in Table 4.

Reviewer 3 Report

1. The abstract needs to be more concise, not what you did but what you concluded.

2. There are many writing errors in the text, please check and correct the full text carefully.

3. The paragraph "Conclusion" is missing in the text, please add it.

Author Response

Comment1 Answer : As the sample size required for multiple logistic regression analysis, we calculated that the independent variables 6 x 10 = 60 people, and the number of study subjects was about 60, it was slightly difficult to obtain the required sample size.

Comment2 Answer : We have clarified the relationship between subjective symptoms and other symptoms, as well as physiological indices, but we could not keep it within the number of pages published, so we have submitted only the current analysis.

Comment3 Answer : We added the missing "Conclusion" paragraph to the text.

Reviewer 4 Report

In this article Sakamoto A and Ikeda M have tried to show that the use of blood flow measurement in limbs in addition to the objective and subjective symptom assessment should be used by nurses in clinical practice to assess the risk of arterial obstruction in patients with type 2 diabetes mellitus.  Overall, the authors have provided a good background information and discussion. Few comments/ suggestions are listed below:

·         Did the authors take into account the history of diabetes control in the patients while including them or excluding them from the study, as the patients with poor diabetes control are more likely to develop disease related complications?

·         Do the authors think that past medication/ medication history of the patients and/ or their treatment strategy for management of diabetes could impact the study findings? Please comment.

·         The authors are recommended to include the study limitations in the discussion section of the manuscript.

Author Response

Comment1 Answer : Patients with inadequate diabetes control are more likely to develop disease-related complications, but since the survey was conducted without a constant degree of diabetes control, it is undeniable that there is a bias in diabetes control. We need to further investigate the progression of arteriosclerosis in the lower extremities under the same conditions of diabetes control in the future.

Comment2 Answer : The data was collected as basic information, considering that the history of ulcers and the history of medications for treatment purposes can affect the results of the study, but the impact could not be clarified and described in this study.

Comment3 Answer : We have added the limitations of the study, including the points you raised.